# Safety and Tolerability of Stromal Vascular Fraction Combined with β-Tricalcium Phosphate in Posterior Lumbar Interbody Fusion: Phase I Clinical Trial

**DOI:** 10.3390/cells9102250

**Published:** 2020-10-08

**Authors:** Kyoung-Tae Kim, Kwang Gi Kim, Un Yong Choi, Sang Heon Lim, Young Jae Kim, Seil Sohn, Seung Hun Sheen, Chan Yeong Heo, Inbo Han

**Affiliations:** 1Department of Neurosurgery, School of Medicine, Kyungpook National University, Daegu 41944, Korea; nskimkt7@gmail.com; 2Department of Neurosurgery, Kyungpook National University Hospital, Daegu 41566, Korea; 3Department of Biomedical Engineering, College of Medicine, Gachon University, Seongnam-si 13120, Korea; kimkg@gachon.ac.kr (K.G.K.); smion123@naver.com (S.H.L.); youngjae@gachon.ac.kr (Y.J.K.); 4Department of Health Sciences and Technology, Gachon Advanced Institute for Health Sciences and Technology (GAIHST), Gachon University, Seongnam-si 13120, Korea; 5Department of Neurosurgery, CHA Bundang Medical Center, CHA University, Seongnam-si 13496, Korea; nschoiuy@gmail.com (U.Y.C.); sisohn@hanmail.net (S.S.); nssheen@gmail.com (S.H.S.); 6Department of Plastic and Reconstructive Surgery, Seoul National University Bundang Hospital, Seongnam-si 13620, Korea; lionheo@gmail.com

**Keywords:** spinal stenosis, posterior lumbar interbody fusion, stromal vascular fraction, bone graft substitute

## Abstract

The rates of pseudarthrosis remain high despite recent advances in bone graft substitutes for spinal fusion surgery. The aim of this single center, non-randomized, open-label clinical trial was to determine the feasibility of combined use of stromal vascular fraction (SVF) and β-tricalcium phosphate (β-TCP) for patients who require posterior lumbar interbody fusion (PLIF) and pedicle screw fixation. Two polyetheretherketone (PEEK) cages were inserted into the intervertebral space following complete removal of the intervertebral disc. The PEEK cage (SVF group) on the right side of the patient was filled with β-TCP in combination with SVF, and the cage on the left side (control group) was filled with β-TCP alone. Fusion rate and cage subsidence were assessed by lumbar spine X-ray and CT at 6 and 12 months postoperatively. At the 6-month follow-up, 54.5% of the SVF group (right-sided cages) and 18.2% of the control group (left-sided cages) had radiologic evidence of bone fusion (*p* = 0.151). The 12-month fusion rate of the right-sided cages was 100%, while that of the left-sided cages was 91.6% (*p* = 0.755). Cage subsidence was not observed. Perioperative combined use of SVF with β-TCP is feasible and safe in patients who require spinal fusion surgery, and it has the potential to increase the early bone fusion rate following spinal fusion surgery.

## 1. Introduction

Spinal fusion surgery is an established surgical technique that fuses two or more vertebrae to achieve spinal stability in various spinal diseases including spinal stenosis, instability, progressive deformity, and spinal fracture [1,2,3,4,5]. As the number of spinal fusion surgeries has increased along with the aging population, the rate of pseudoarthrosis (failure of bony fusion) has rapidly increased and has been reported to be as high as 48% [1,4,6,7]. To increase the spinal fusion rate, various methods, including autologous or allogenic bone graft, demineralized bone matrix (DBM), ceramics, bone morphogenetic proteins (BMPs), autologous growth factors, and synthetic peptides, have been developed and used [6,8,9,10,11,12]. Despite the remarkable advancement in the field of bone graft substitutes for spinal fusion, there is no graft substitute/expander that has demonstrated a clear superiority over autogenous iliac crest bone graft, and an ideal bone graft substitute with equal or superior efficacy combined with minimal complications does not seem to exist at this time[10].

Previous studies have shown that mesenchymal stem cells (MSCs) from bone marrow or adipose tissue may hold great promise in repairing musculoskeletal tissues [12,13,14,15]. Compared to bone marrow, adipose tissue is a better source of MSC isolation due to easy isolation, less donor site morbidity, and large amount of cell numbers. Stromal vascular fraction (SVF) isolated from adipose tissue is a heterogeneous cell population obtained after collagenase treatment of adipose tissue and includes adipose-derived MSCs (ASCs) [16]. Adipose-derived MSCs (ASCs) is well known to induce bone regeneration in preclinical study [17,18]. For clinical study, however, isolated ASCs should be expanded through in vitro culture and approved by a government regulatory institute. Furthermore, some studies showed that SVF had better osteoinductive capabilities than ASCs [18]. Processing for SVF isolation is not difficult, and SVF containing heterogenous cell populations such as stem cells and endothelial cells could be isolated within 90 min [16]. There have been encouraging results for the application of SVF in bone regeneration [19,20,21,22,23]. Thus, strong attention has been paid to the SVF, non-cultured fraction of ASCs due to an easier cell management preparation for clinical application.

For several decades, posterior lumbar interbody fusion (PLIF) using double cages has been accepted as the standard treatment for lumbar degenerative disc disease requiring surgery [24]. Advantages of the PLIF procedure include restoration of disc height, disc stabilization, nerve root decompression, and reinforcement of the weight-bearing axis in the anterior segment of the spinal column [25]. The PLIF procedure achieves spinal fusion by inserting single or double cages made of either allograft bone or synthetic materials (polyetheretherketone (PEEK) or titanium) directly into the disc space following complete removal of the intervertebral disc. PEEK cage is a non-absorbable biopolymer and is packed with various materials including autologous bone grafts, demineralized bone graft, or synthetic bone grafts. PEEK cages have been widely used and accepted, with excellent surgical outcomes for spinal fusion surgery [24,25]. The use of interbody cages may enhance better long-term fusion stability. However, pseudarthrosis is still one of the most challenging problems.

In this study, we determined the feasibility of β-tricalcium phosphate (β-TCP)/SVF mixture to accelerate bony fusion following PLIF surgery. We used an automated system for SVF isolation from adipose tissue to strictly control technical procedures. We performed a single center, non-randomized, open-label phase I clinical trial comparing the fusion rate following PLIF surgery using two PEEK cages filled with different materials (a cage on the right side of the patient with a β-TCP/SVF mixture versus a cage on the left side with β-TCP alone).

## 2. Patients and Methods

This study protocol was approved by the Institutional Review Board of CHA University CHA Bundang Medical Center (approval number: CHAMC 2018-01-032). Written informed consent was obtained from each patient included in this study. This study was performed in full accordance with the principles of the Declaration of Helsinki.

### 2.1. Patients

The study was performed by using a prospective study design with 10 patients who had undergone PLIF and pedicle screw fixation procedure at Bundang CHA Medical Center within the period of June 2018 to March 2020. PLIF and pedicle screw fixation are widley accepted surgical techniques for the management of severe lumbar spinal stenosis and spondylolisthesis with instability. The major inclusion criteria were male or female between the ages of 19 and 75. All patients were diagnosed with lumbar disc herniation, spinal stenosis, degenerative, or spondylolytic spondylolisthesis requiring PLIF surgery at one or two segments. The diagnosis of lumbar stenosis was based on clinical findings and MRI. Patients could be diagnosed with lumbar stenosis due to neurotic intermittent claudication and/or nerve root compression by MRI. All patients had significant low back pain and leg pain unresponsive to active non-surgical treatments for at least 6 months. Patients were excluded if they had any infection, spinal fracture, or other disorders including autoimmune diseases, malignancies, or hemorrhagic diseases. Patients were also excluded if they were receiving corticosteroids, immunosuppressive drugs, or chemotherapy. Those who participated in other clinical trials within 30 days of the trial start were also excluded.

### 2.2. SVF Isolation with an Automated System

The Cellunit SVF Isolation System (CGBio Inc., Seongnam, Korea) was used for automated SVF isolation from aspirated adipose tissue in this study (Figure 1). In brief, abdominal adipose tissues were aspirated using a 50-mL syringe attached to a cannula with a 3-mm inner diameter following general anesthesia. The average amount of aspirated fat tissue obtained was 46.1 ± 6.4 mL (range, 40 to 60 mL). Aspirated fat tissue was transferred into the chamber and enzymatically digested with 0.1% collagenase (SERVA Electrophoresis GmbH, Heidelberg, Germany). After digestion, the solution was washed with saline to remove collagenase and centrifuged to obtain SVF cells in the same chamber. SVF cells were collected within 45 min. The final cell output was approximately 80 mL SVF cell suspension. For the administration of SVF, the additional centrifugation was performed to concentrate the SVF cell suspension into 5 mL. The number of nucleated SVF cells was determined using an automated cell counter (Luna-FLTM; Logos Biosystems, Anyang). The total amount of injected SVF cell suspension was 1.6 ± 1.2 ×10^7^ cells (range, 0.3 to 3.7 × 10^7^ cells) at a final concentration of 2.7 ± 1.8 × 10^5^ cells/mL (range, 0.5 to 6.2 ×10^5^ cells/mL). The cell viability of isolated SVF cells was 83.4 ± 9.7 % (range, 69 to 100 %) (Table 1).

### 2.3. Spinal Fusion Surgery and SVF Implantation

We performed PLIF augmented with pedicle screw fixation (PSF) combined with SVF cell injection in a total of 10 patients with spinal stenosis. All surgeries were performed by the same surgeon. The SVF was isolated from aspirated adipose tissue processed with the automated cell isolation system (Figure 1). Briefly, the fat tissue for SVF isolation was first harvested by liposuction under general anesthesia. During the SVF processing, the patients were positioned in the prone position and underwent PLIF augmented with PSF. After subtotal laminectomy (subtotal removal of the lamina portion of the bone) and complete removal of the intervertebral disc, two PEEK cages packed with injectable β-TCP/SVF mixture (the cage on the right side of the patient) or β-TCP alone (the cage on the left side) were inserted in the same disc space. Pedicle screws were then placed in the upper and lower vertebral bodies to provide immediate stability. For the PLIF cage on the left side, 1 mL β-TCP bone substitute was mixed with 1 mL normal saline (1:1 ratio) using a syringe connector until the materials became a smooth paste. The β-TCP bone substitute/saline mixture was mixed with autologous bone chips obtained from laminectomy at 1:1 ratio. For the PLIF cage on the right side, an injectable β-TCP bone substitute of 1 mL was mixed with 1 mL SVF cell suspension (1:1 ratio) using a syringe connector. The β-TCP bone substitute/SVF mixture was mixed with autologous bone chips at a 1:1 ratio. To prevent SVF and β-TCP from escaping from the cages, we used PEEK cages without lateral side holes (Figure 1).

### 2.4. Evaluation of Safety and Efficacy

Clinical outcomes were assessed using the visual analogue scale (VAS) of low back pain and leg pain. Spine X-ray and computed tomography (CT) were performed preoperatively and again at 6 and 12 months after surgery to evaluate bone fusion inside the cage. To compare the fusion rate, we cropped the operated intervertebral space from the coronal and sagittal CT images, visualizing the center portion of the PEEK cages. Fusion status was ranked using 5 grades according to the Brantigan–Steffee classification[26] (Table 2). In the Brantigan–Steffee classification system, grades 1, 2, and 3 were defined as the non-union state, and grades 4 and 5 were defined as the union (bone fusion) state. Additionally, we evaluated fusion bone quality using the Housfield unit (HU) measured by CT (Figure 2). CT images of the spine from the 6- and 12-month follow-up were used to compare the bone formation between β-TCP/SVF mixture and β-TCP alone inside interbody fusion cages. For feature analysis, first-order and second-order features (Gray Level Cooccurrence Matrix, GLCM) were extracted from spine CT. In total, 19 first-order features and 22 GLCM features were extracted from our data [27]. Since the HU value in CT is related to bone mineral density (BMD), the feature value was extracted based on the HU value in the region of interest [28].

For the safety study, information on adverse events such as infection, cage subsidence, and persistent or recurrent back and leg pain was collected at each visit (1, 3, 6, 9, and 12 months following surgery). Cage subsidence was evaluated using postoperative serial follow-up lateral radiographs and was defined to be present if a cage was observed to migration into an adjacent vertebral body by ≥2 mm based on comparisons with previous radiographs at each visit (1, 3, 6, 9, and 12 months following surgery) [29]. Safety metrics were also monitored at each visit by evaluating simple radiography, laboratory findings, and vital signs.

### 2.5. Statistical Analysis

All data were expressed as the mean ± standard error of mean (SEM). The fusion rates between the right PLIF cage and the left PLIF cage were compared, and they did not follow a normal distribution, so a Mann–Whitney U test was used as a statistical analysis method. Since the comparative analysis of HU values follows a normality distribution, each feature value was analyzed with incremental reduction analysis and an independent samples *t*-test. The incremental reduction analysis calculated the difference between the mean feature values at follow-up, and the *t*-test compared the left and right differences. A *p*-value < 0.05 was considered statistically significant. All statistical analyses were performed using SPSS for Windows version 25.0 (SPSS, Inc., Chicago, IL, USA).

## 3. Results

### 3.1. Baseline Characteristics

This studied included 10 patients (4 males and 6 females) with a mean age of 64.4 ± 7.1 years (range, 52 to 75 years) who were recruited at the Spine Center of CHA Bundang Medical Center between July 2018 and March 2019 (Table 1). A total of 12 PLIF procedures were performed. Two patients had 2 segment interbody fusions and 8 patients had 1 segment fusion. The mean cell number in SVF was 2.7 × 10^5^ cells/mL, and the viability of the nucleated SVF cells was 83.4 ± 9.7 %. The main characteristics of the patients are summarized in Table 1. No patient was lost to follow-up during the one-year study.

### 3.2. Clinical and Radiographic Outcomes

Pain assessments with VAS were performed preoperatively and at 1, 3, 6, and 12 months follow-up. VAS scores were reduced from a mean of 7.9 ± 1.3 preoperatively to 4 ± 0.9 at 1 month, 3.2 ± 1.1 at 3 months, and 2.7 ± 1.6 at 12 month follow-ups, respectively (Table 1).

In the fusion grade assessment using the CT at 6 months, the fusion grades of the cages were as follows: the right side (SVF group, cages filled with SVF/β-TCP mixture) was 3.6 and the left side (control cages filled with β-TCP alone) was 2.8. The difference between the two groups was statistically significant (*p* = 0.017; Table 2; Figure 3, Figure 4 and Figure 5). The CT assessment of fusion at 6 months postsurgery showed a 54.5% fusion rate in the SVF group (cages filled with SVF and β-TCP mixture) and an 18.1% fusion rate in the control cages (cages filled with β-TCP alone) (*p* = 0.151). On the CT performed at 12 months, the fusion grade was 4.5 in cages filled with SVF and β-TCP mixture and 4.1 in the control cages filled with β-TCP alone (*p* = 0.06). Additionally, a 100% fusion rate was observed in the cages filled with the SVF and β-TCP mixture, while the fusion rate in the control cages was 91.7%, and no statistical difference (*p* = 0.755) was found. In the cages filled with the SVF and β-TCP mixture, a tendency to fuse faster than the cages filled with β-TCP alone was observed, although there was no statistical difference. Statistical analysis according to sex showed no difference in fusion rate and fusion grade.

In the HU analysis for the evaluation bone formation inside an interbody cages, the maximum value was higher in the cages filled with β-TCP/SVF mixture than the cages with β-TCP alone in both the data collected in the 6th month and the 12th month, though the difference was greater in the 12th month (Figure 6A). The mean was not significantly different between the cages filled with β-TCP/SVF mixture and the cages with β-TCP alone (Figure 6B). We presented the average value of the features of each group in Table 3. There was no significant difference in mean values, but the maximum and range tended to increase in the experimental group. Although the minimum and the mean did not show a significant difference, the increase in the maximum and the range indicated that the bone inside the region of interest (ROI) was partially formed.

### 3.3. Safety Outcome

Adverse events were observed in 3 out of 10 patients (Table 4). Patient No. 2 patient was observed to have early gastric cancer in a health medical examination gastrointestinal endoscopy performed 5 months after surgery. Subsequently, the patient underwent gastric submucosal dissection and had continuous follow-up in the outpatient clinic after recovery. Patient No. 4 showed no abnormality on a chest radiograph on the preoperative examination, but a 1.5 cm size nodule was observed on the right middle lobe on chest CT performed at the health medical examination conducted 3 months after surgery. Subsequently, the nodule size gradually decreased on serial CT, and the inflammatory lesion was diagnosed and is being continuously followed up in the outpatient clinic. A 38-degree fever was observed in Patient No. 9 on the 4th day after surgery, and staphylococcus epidermidis was cultured in the blood culture. The patient was subjected to reoperation (grade 4 by the CTCAE scale), irrigation, and debridement, and then the patient completely recovered after intravenous antibiotic treatment and was discharged 30 days after the first surgery. Two of the adverse events mentioned above, except for surgical site infection, were definitely not related to this study.

## 4. Discussion

Cell-based therapy has recently attracted attention for its potential to enhance the fusion rate in spinal fusion surgery, and there is a report on the clinical use of cell-based products for lumbar spinal fusion [30]. However, there is currently little data on cell-based products [30]. Cell-based allografts maintain their native bone-forming cells, such as osteoprogenitor cells and MSCs, along with bone matrix components [10]. It has been reported that bone marrow aspirates (BMAs) combined with a collagen sponge or ceramic resulted in similar fusion rates compared to local bone, whereas BMAs were reported to have lower fusion rates in comparison to iliac bone grafts [31]. β-TCP is a synthetic bone substitute that combines type 1 collagen and tricalcium phosphate with a highly porous scaffold that supports bone growth [32,33,34]. β-TCP is typically augmented by BMAs to add osteogenic and osteoinductive qualities [34,35]. Gan Y et al. used enriched bone marrow-derived MSCs harvested from the bilateral iliac crest in conjunction with β-TCP for posterior spinal fusion in 41 patients and reported a 95.1% spinal fusion rate at 34.5 months [36]. In that study, the enriched MSCs were created by a cell processor perioperatively and were subsequently implanted back into the patient. Mc Anany et al. analyzed 57 patients who underwent a one- or two-level instrumented anterior cervical discectomy and fusion (ACDF) procedure using interbody allograft and Osteocel™ (NuVasive, San Diego, CA, USA) [37]. Osteocel™ is known to be a viable bone matrix product that preserves the native stem cells found in marrow rich bone. At the 1-year follow-up, 87.7% of the patients in the Osteocel cohort showed solid fusion compared with 94.7% in the control group (*p* = 0.19) [37]. In another study, Vanichkachorn J et al. reported a radiological fusion rate of 93.5% at 12 months follow-up in 31 patients undergoing single-level ACDF [38]. Eastlack RK et al. evaluated the use of Osteocel Plus™ cellular allograft for the ACDF of 249 levels in 182 patients [39]. In subjects treated at a single level with a minimum of 24-months follow-up, 92% of levels demonstrated solid bridging [39]. In combined single- and two-level procedures, 87% of levels demonstrated solid fusion at 24 months [39].

In this study, 10 patients who underwent PLIF with PSF were analyzed for the bone regenerative potential of SVF. After total removal of the disc, two PEEK cages filled with different mixtures were inserted into the intervertebral disc space. The right-sided cage was filled with autologous SVF and β-TCP, and the left-sided cage was filled with β-TCP alone. We found that newly-isolated adipose-derived SVF cells, when implanted directly into a bone fusion setting, can engender further bone formation and augment bone fusion. Originally demonstrated in an animal model, this is the first human clinical trial on this concept for enhanced spinal fusion. The current study design was a phase I clinical trial, and calculating an adequate sample size can be problematic. It was expected that 10 patients would be sufficient for the trial to achieve the trial endpoints as the active sample size in phase I clinical trials is standardly reported to be from six to 10 active subjects [40,41]. Following this study design protocol, we obtained the following main findings: combined implantation of SVF and β-TCP is safe and tolerable for PLIF surgery; additionally, at 6 months following surgery, 54.5% of the cages filled with SVF and β-TCP (right-sided cages) and 18.2% in cages filled with β-TCP alone (left-sided cages) had radiologic evidence of bone fusion (*p* = 0.151). The 12-month fusion rate of the right-sided cages was 100%, while that of the left-sided cages was 91.6% (*p* = 0.755). Cage subsidence was not observed in any cage.Newly-isolated SVF can regenerate bone tissue faster compared to expanded ASCs, despite SVFs having a reduced number of ASCs [38]. One possible reason is that different cell populations contained in SVF probably work together to better engender MSC activity compared to ASCs alone, which may suggest a central interaction between ASCs and the small-scale environment [17]. Furthermore, previous studies have reported that human ASCs closely work together with vascular endothelial cells and release growth factors and cytokines into the extracellular environment, such that it affects different regions in the human body [42,43]. Among them, VEGF [44,45,46], a paracrine factor secreted by ASCs, can increase the differentiation of osteoblast. VEGF is an essential factor in skeletal growth and a central mediator in angiogenesis. Previous studies have shown that SVF engenders angiogenesis and provides positive effects on small-scale circulation along with increasing VEGF levels in the blood serum [44,45]. This engenders tissue growth and repair by augmenting and reshaping the blood vessel network; furthermore, it crucially plays a role in bone repair and growth at a systemwide level [47]. Equally, BMP-2 is connected to the modulation of osteogenic differentiation [48]. SVF cells can dramatically increase osteogenic-specific marker expression through the modulation of BMP-2 and/or TGF-β, and SVF increases osteogenesis potential and improves osteoinduction [49].

SVF has the same role with BMAs, so the combination of SVF and β-TCP may have a synergistic effect in our study [34,35]. Prins HJ et al. demonstrated the feasibility and potential efficacy of SVF seeded on bone substitutes for maxillary sinus floor elevation and autologous SVFs with calcium phosphate ceramics showed increased maxiallry bone height for dental implantations [50]. For each maxillary sinus, they prepared 2 g calcium phosphate carrier with 20 × 10^6^ cells (2.67 mL of the 7.5 × 10^6^ cells per milliliter) and showed potential efficacy of SVF for bone regeneration [50]. SVF cells can allow for good graft substitution in conjunction with an appropriate carrier. To engraft artificial bone, cells need to be attached closely to the artificial bone matrix. Bone formation at the base of the implant is crucial. β-TCP is non-cytotoxic and has strong biocompatibility; it can result in the most total bone volume in comparison with other grafts [51,52,53]. In particular, the SVF has been shown to have good affinity, proliferation, and osteogenic differentiation on β-TCP, and secrete a high number of growth factors [54,55]. However, β-TCP can have a low bone growth rate, which can be due to a lack of osteo-inductive potential along with only bone conduction properties [50]. Therefore, we injected the PEEK cages with β-TCP in combination with SVF. The precise mechanism underlying the combined implantation of SVF and β-TCP that led to spinal fusion improvement in the present study remains unclear. Based on these studies, however, we assume that combined implantation of SVF and β-TCP improves spinal fusion.

Our results showed a higher grade of intervertebral fusion status assessed by CT scans 6 months postsurgery. Additionally, the maximum value of HU analysis was higher in the cages filled with SVF/β-TCP mixture than the cages filled with β-TCP alone in both the data collected at the 6th month and the 12th month, though the difference was greater in the 12th month (Figure 6). This is demonstrated by the influence of angiogenesis, stimulation, and transformation of fibroblasts, and mobilization of endogenic stem cells, which are key functions of ASCs included in SVF, and through the production of cytokines and growth factors, through paracrine stimulation of SVF [56]. Our study is a single-arm, open-label, phase I pilot study, and thus, caution should be applied when drawing any conclusions regarding long-term safety and efficacy. In addition, our study has limitations including a small number of participants, an absence of control subjects, and a wide range of SVF cell number and cell viability (cell number: 0.5 to 6.2 ×10^5^ cells/mL; cell viability: 69 to 100%). In all 10 patients, β-TCP plus SVF were implanted into the right-side cage, whereas β-TCP only was implanted into the left-sided cage. This allowed intrapatient evaluation of the potential added value of SVF supplementation. In our study, the amounts of liposuction (range, 40 to 60 mL) and SVF cell suspension (2.7 ± 1.8 ×10^5^ cells/mL) were relatively less than that of the previous study [50]. The reason for the wide range and the small number of SVF cells is that a relatively small amount of liposuction was harvested compared to previous studies, and the amount of collection varied from patient to patient. Thus, a large-scale clinical trial is necessary to assess optimal SVF cell dose and optimal SVF carrier that will benefit from SVF co-implantation to promote spinal fusion. However, we propose that co-administration of SVF and β-TCP into the cages may provide a safe and tolerable treatment for enhancing spinal fusion.

## 5. Conclusions

This is the first-in human study using freshly isolated, autologous SVF appled in a one-step surgical procedure with β-TCP to increase fusion rate in patients requiring PLIF surgery. This phase I clinical study demonstrated the safety, feasibility, and potential efficacy of SVF seeded on bone substitutes such as β-TCP, providing new insight to offer broad potential for SVF application in regenerative medicine for spinal disorders.

## Figures and Tables

**Figure 1 cells-09-02250-f001:**
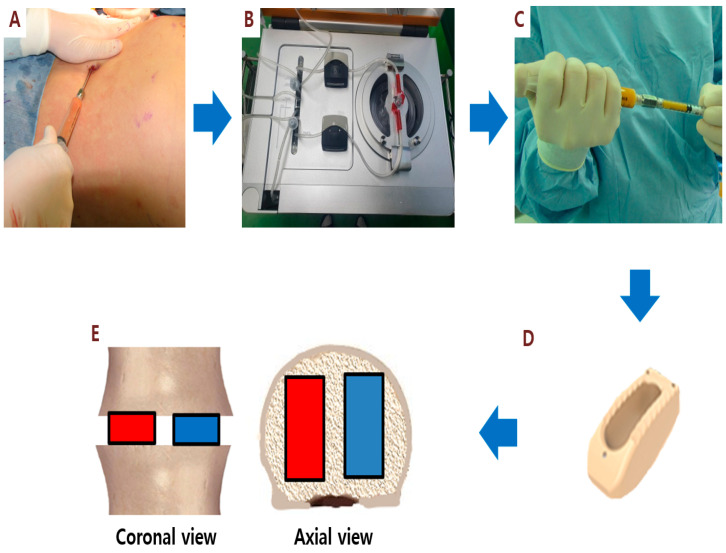
Stromal vascular fraction (SVF) isolation and injection. Step 1: Adipose tissue is harvested by manual liposuction (**A**). The Cellunit^®^ System (**B**) generates adipose-derived SVF (**C**) from adipose tissue collected manually. For posterior lumbar interbody fusion, the polyetheretherketone (PEEK) cage without lateral side holes (**D**) was used. The PEEK cage on the right side (red color) of the patient is packed with SVF and β-tricalcium phosphate (β-TCP) mixture and the cage on the left side (blue color) is packed with SVF alone(**E**).

**Figure 2 cells-09-02250-f002:**
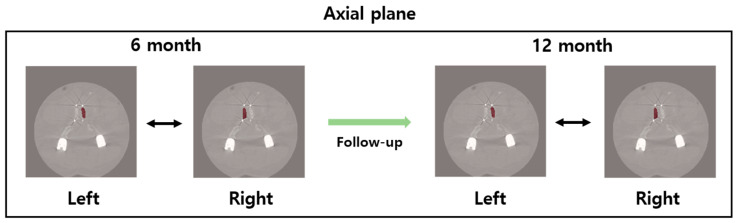
The follow-up spine CT for feature analysis. Overlay of red color: Region of Interest of spine.

**Figure 3 cells-09-02250-f003:**
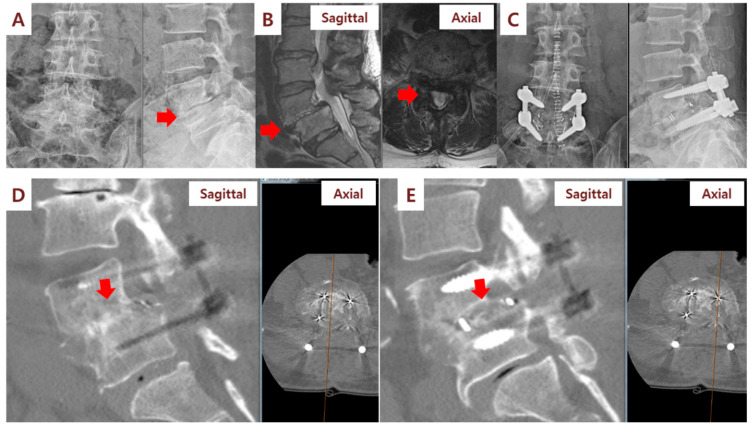
Illustration case (Patient No. 1): A 60-year-old woman received posterior lumbar interbody fusion and pedicle screw fixation. Preoperative lumbar spine standing anteroposterior and lateral view (**A**) and magnetic resonance imaging (MRI) (**B**) demonstrating L4/L5 severe spinal stenosis with degenerative spondylolisthesis. (**C**) Postoperative lumbar spine standing anteroposterior and lateral view. (**D**) Computerized tomography (CT) sagittal view performed at 6 months after surgery showed fusion status (red arrow) of cages filled with SVF and β-TCP mixture on the right side of the patient. (**E**) CT sagittal view showed no fusion status (red arrow) of cages filled with β-TCP alone on the left side.

**Figure 4 cells-09-02250-f004:**
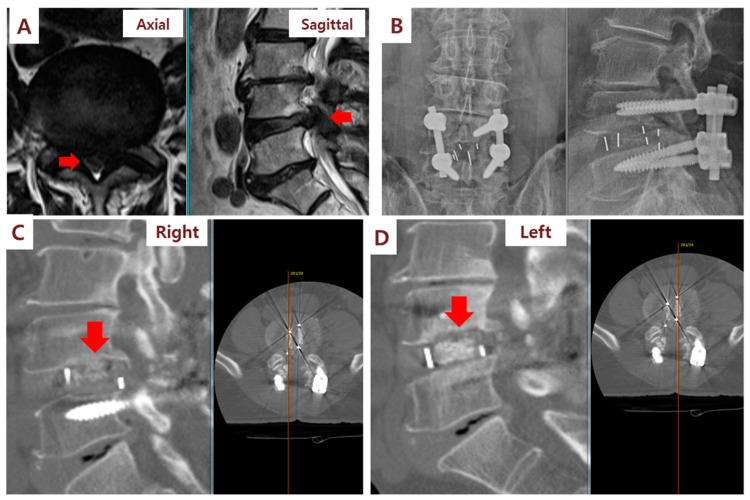
Illustration case (Patient No. 3): A 67-year-old male received posterior lumbar interbody fusion and pedicle screw fixation. (**A**) Preoperative lumbar axial and sagittal MRI demonstrating L4/L5 severe spinal stenosis. (**B**) Postoperative lumbar spine standing anteroposterior and lateral view. (**C**) CT sagittal view at 6 months after surgery showed fusion status (red arrow) of cages filled with SVF and β-TCP mixture on the right side of the patient. (**D**) By contrast, CT sagittal view showed no fusion status (red arrow) of cages filled with β-TCP alone on the left side.

**Figure 5 cells-09-02250-f005:**
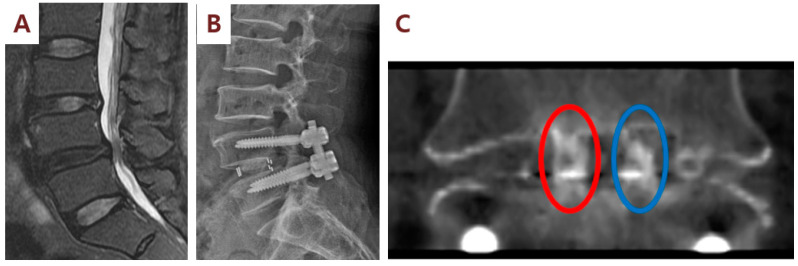
Illustration case (Patient No. 3): A 63-year-old male received posterior lumbar interbody fusion and pedicle screw fixation. (**A**) Preoperative lumbar sagittal MRI demonstrating L4/L5 severe spinal stenosis. (**B**) Postoperative lumbar spine standing lateral view. (**C**) CT coronal view at 6 months after surgery showed the fusion status of cages (red eclipse) filled with SVF and β-TCP mixture on the right side of the patient. By contrast, CT coronal image showed no fusion status (blue eclipse) of cages filled with β-TCP alone on the left side.

**Figure 6 cells-09-02250-f006:**
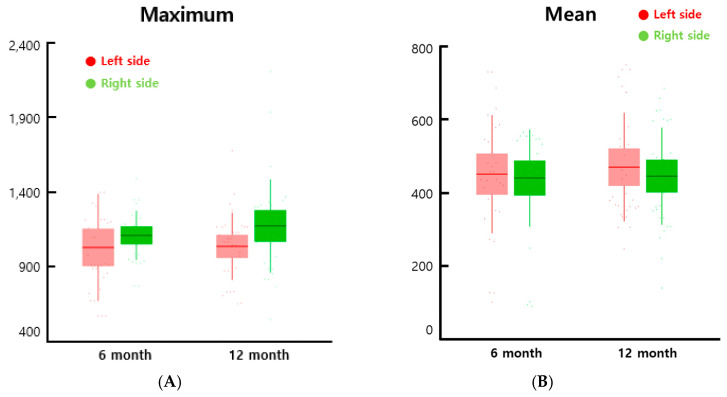
Maximum (**A**) and mean feature (**B**) values extracted from the axial plane of spine CT. The left and right sides are represented by red and green bars. The middle line is the mean value, the bar is the standard deviation, and the points are the feature values. The maximum is the largest Housfield unit (HU) value, and the mean is the average gray level intensity inside the region of interest (ROI).

**Table 1 cells-09-02250-t001:** Demographic characteristics of patients.

	Mean	SD
Age (year)	64.4	7.2
Sex (M:F)	4:6	
BMI (kg/m^2^)	25.5	3.0
BMD (g/cm^2^)	0.84	0.12
HTN	6	
DM	5	
Smoking	3	
preVAS	7.9	1.3
Fusion level		
L3/4	3	
L4/5	8	
L5/S1	1	
Live cells/mL (10^5^)	2.7	1.8
Viability (%)	83.4	9.7

**Table 2 cells-09-02250-t002:** Assessment of fusion grade by Brantigan–Steffee classification.

Patient Number	Surgical Level	6 Month Right	6 Month Left	12 Month Right	12 Month Left
1	L4/5	4	3	5	5
2	L4/5	3	2	5	4
3	L4/5	4	3	5	3
4	L3/4	4	3	5	4
4	L4/5	3	2	4	4
5	L4/5	4	3	5	4
6	L4/5	5	4	5	5
7	L5/S1	4	2	4	4
8	L3/4	3	3	4	4
9	L3/4	3	3	5	4
9	L4/5	3	2	4	4
10	L4/5	4	4	4	4

**Table 3 cells-09-02250-t003:** The result of radiomic feature analysis.

Feature	Statistic 6-Month (LS vs. RS)	Statistic 12-Month (LS vs. RS)	Average of Feature Value (Control)	Average of Feature Value (Experimental)
Maximum	*p*: 0.92 df: 63.00 sd: 271.92	*p*: 0.33 df: 62.00 sd: 232.18	6 m: 978.84 12 m: 1052.47	6 m: 985.42 12 m: 1110.09
Mean	*p*: 0.62 df: 63.00 sd: 165.54	*p*: 0.88 df: 63.00 sd : 140.72	6 m: 484.37 12 m: 504.71	6 m: 473.01 12 m: 478.36
Minimum	*p*: 0.91 df: 63.00 sd: 333.32	*p*: 0.56 df : 63.00 sd : 306.67	6 m: −156.03 12 m: −146.941	6 m: −308.6 12 m: −353.47
Range	*p*: 0.98 df: 63.00 sd: 459.07	*p*: 0.35 df: 63.00 sd: 437.25	6 m: 1134.88 12 m: 1361.07	6 m: 1132.36 12 m: 1463.56

*p*: *p*-value; df: degrees of freedom; sd: pooled standard deviation.

**Table 4 cells-09-02250-t004:** Summary of adverse events.

Patient Number	Adverse Event	Treatment-Related	Unanticipated Problem	NCI-CTCAE Scale
2	Early Gastric Cancer	Definitely not related	Unexpected	3
4	Solitary pulmonary nodule	Definitely not related	Unexpected	1
9	Surgical site infection	Possible not related	Unexpected	4

NCI-CTCAE : National Cancer Institute Common Terminology Criteria for Adverse Events.

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
