# Peer review of "Safety and Tolerability of Stromal Vascular Fraction Combined with β-Tricalcium Phosphate in Posterior Lumbar Interbody Fusion: Phase I Clinical Trial"

_cells, 2020, doi:10.3390/cells9102250_

Round 1
Reviewer 1 Report
This is a phase 1 clinical trial that is an important step in the development of approaches to spinal fusion. My main concern with this manuscript is that a large part of the manuscript gives the impression that this is an efficacy study as opposed to a phase 1 trial for the evaluation of safety and tolerability. The title/abstract/primary stated aim of the study should be modified to reflect that this is a phase 1 trial and the conclusions should focus on this modified aim. The study's conclusions regarding efficacy are also not supported by many of the results because they were not statistically significant and with a phase 1 trial it is very difficult to make assumptions about the clinical significance of the results. My other suggestions to help improve the manuscript are:
- A number of statements throughout the manuscript require references
- 2.1 Please provide information relating to the recruitment strategy
- 2.1 I am uncertain why the inclusion criteria included people as young as 19 when the target population had degenerative spinal stenosis
- 2.1 Please elaborate on how spinal stenosis meeting inclusion criteria was established
- 2.2 How was the dosage decided? Were these standard doses? Provide references if they are.
- 2.4 How was cage subsidence assessed at each visit?
- 2.5 Please explain why a combination of Mann-Whitney U tests and t tests were used for analysis
- 4 The first 2 paragraphs of the discussion appear to be better suited to the introduction.
Author Response
Overall Comment:
This is a phase 1 clinical trial that is an important step in the development of approaches to spinal fusion. My main concern with this manuscript is that a large part of the manuscript gives the impression that this is an efficacy study as opposed to a phase 1 trial for the evaluation of safety and tolerability. The title/abstract/primary stated aim of the study should be modified to reflect that this is a phase 1 trial and the conclusions should focus on this modified aim. The study's conclusions regarding efficacy are also not supported by many of the results because they were not statistically significant and with a phase 1 trial it is very difficult to make assumptions about the clinical significance of the results. My other suggestions to help improve the manuscript are:
RESPONSE: Thank you for pointing this out. We agree with the reviewer’s comment. As suggested by the reviewer, we modified Title, Abstract, and Conclusions.
Line 25-28, Line 36-38, Line 79, 81-82, Line 331-335.
COMMENT 1: A number of statements throughout the manuscript require references.
(Line 238~240)
RESPONSE 1: We thank for the reviewer’s comment. Based on the suggestion, we added refences to the statements of our manuscript.
COMMENT 2: Please provide information relating to the recruitment strategy
RESPONSE 2: Thank you for pointing this out. We have added the recruitment strategy.
(Line 91~98)
The study performed by using prospective study design with 10 patients who had undergone PLIF and pedicle screw fixation procedure at Bundang CHA Medical Center within period June 2018 to March 2020. All patients had significant low back pain and leg pain unresponsive to active nonsurgical treatments for at least 6 months.
COMMENT 3: I am uncertain why the inclusion criteria included people as young as 19 when the target population had degenerative spinal stenosis
RESPONSE 3: We appreciate the reviewer’s concern. The lumbar spine can be fused by means of PLIF and pedicle screw fixation. Indications for PLIF and pedicle screw fixation include severe ruptured disc causing cauda equine syndrome, severe spinal stenosis, degenerative or spondylolytic spondylolisthesis causing nerve root compression. Even young people can suffer from neurologic deficit resulting from cauda equine syndrome. So we include people older than 19 years. In Republic of Korea, 19 years old is legally considered as an adult.
COMMENT 4: Please elaborate on how spinal stenosis meeting inclusion criteria was established
RESPONSE 4: Thank you for pointing this out. The diagnosis of lumbar spinal stenosis was based on clinical findings and MRI. The patients could be diagnosed as lumbar spinal stenosis due to severe neurogenic intermittent claudication and cauda-equina compression and/or nerve root compression by MRI. (Line 97-100).
COMMENT 5: How was the dosage decided? Were these standard doses? Provide references if they are.
RESPONSE 5: Thank you for pointing this out.
In this study, the average amount of aspirated fat tissue obtained was 46.1 ± 6.4 ml (range, 40 to 60 ml) and the average amount of SVF used was 2.7 ± 1.8 × 105 cells. According to one paper studied by Prins HJ, et al. this phase I study evaluated the potential additive effect on bone regeneration by the addition of freshly isolated, autologous but heterologous stromal vascular fraction (SVF). From 10 patients who were partially edentulous in the posterior maxilla and required dental implants for prosthetic rehabilitation, SVF was procured using automatic processing, seeded on either β-tricalcium phosphate (n = 5) or biphasic calcium phosphate carriers (n = 5). For each maxillary sinus, they prepared 2 g calcium phosphate carrier with 20 × 106 cells (2.67 ml of the 7.5 × 106 cells per milliliter). In our study, therefore, the amount of liposuction and SVF cell suspension were relatively less than those of the reference journal. However, our study shows a positive effect on spinal fusion even though we used a relatively small amount of SVF and, so in the next study, more liposuction will be performed and a more consistent amount of SVF will be administered. (Line 323~326)
“The reason for the wide range and the small number of SVF cells is that a relatively small amount of liposuction was harvested compared to previous studies, and the amount of collection varied from patient to patient.”
Reference: Bone Regeneration Using the Freshly Isolated Autologous Stromal Vascular Fraction of Adipose Tissue in Combination With Calcium Phosphate Ceramics; Stem Cells Transl Med. 2016 Oct;5(10):1362-1374. doi: 10.5966/sctm.2015-0369
COMMENT 6: How was cage subsidence assessed at each visit?
RESPONSE 6: Thank you for pointing this out. Patient was evaluated simple radiography, laboratory finding and vital signs at each visit(1,3,6,9 and 12month following surgery, Cage subsidence was evaluated using lateral radiographs and was defined as more than 2-mm migration of the cage into the adjacent vertebral body. (line 156-159)
Reference: Risk Factors of Cage Subsidence in Patients Received Minimally Invasive Transforaminal Lumbar Interbody Fusion; SPINE: October 1, 2020 - Volume 45 - Issue 19 - p E1279-E1285
COMMENT 7: Please explain why a combination of Mann-Whitney U tests and t tests were used for analysis
RESPONSE 7: Thank you for pointing this out. The spinal fusion rate and grade system did not follow the normality distribution, statistical analysis was performed using a nonparametric method through Mann-Whitney U test. For evaluation of HU values follows the normality distribution, so it was analyzed by independent samples t-test, which is a parametric method analysis.(Line 164-168)
“The fusion rates between the right PLIF cage and the left PLIF cage were compared and they did not follow a normal distribution, so a Mann-Whitney U test was used as a statistical analysis method. Since the comparative analysis of HU values follows a normality distribution, each feature value was analyzed with incremental reduction analysis and an independent samples t-test.”
COMMENT 8: The first 2 paragraphs of the discussion appear to be better suited to the introduction.
RESPONSE 8: Thank you for pointing this out. We moved first part of Discussion to Introduction and modified Introduction (Line 50-67).
Reviewer 2 Report
hoi et al. conducted a clinical trial entitled "Clinical Efficacy of Stromal Vascular Fraction Combined with β-Tricalcium Phosphate in Posterior Lumbar Interbody Fusion". It is very important subject but there are several comments and suggestions to improve the study. The major drawback of the study is a small number of participants with a surgery side effects of nearly in the 40% of participants, in addition, to include 4 females in total number leave a very small number to conclude anything. Authers did not also compare the data of male and females. There is an absence of control subjects and a wide range of SVF cell numbers and cell viability as authors mentioned also force me to suggest the inclusion of more participant before conclude anything. There is not discusion of including TCP. No data mentioned of VAS in Table 1 except pretreatment VAS score. Figure legend3- Pls correct the word white arrow in D and E as there is no white arrow. All the MRI and CT images should be lebeled properly.Author Response
Comment 1:
Choi et al. conducted a clinical trial entitled "Clinical Efficacy of Stromal Vascular Fraction Combined with β-Tricalcium Phosphate in Posterior Lumbar Interbody Fusion". It is very important subject but there are several comments and suggestions to improve the study. The major drawback of the study is a small number of participants with a surgery side effects of nearly in the 40% of participants,
RESPONSE 1: Thank you for pointing this out. We performed surgery on 10 patients, of which 3 had adverse events. Among them, except for surgical site infection, it is judged to have definitely no relation to this study. Early gastric cancer and solitary pulmonary nodule are not related to SVF implantation, PLIF and pedicle screw fixation. (Line 232-233)
in addition, to include 4 females in total number leave a very small number to conclude anything.
RESPONSE 2: The current study design was a phase I clinical trial, and calculating an adequate sample size can be problematic. It was expected that 10 patients would be sufficient for the trial to achieve the trial endpoints as the active sample size in phase I clinical trials is standardly reported to be from six to 10 active subjects.(Line 267-271)
Authors did not also compare the data of male and females.
RESPONSE 3: We appreciate the reviewer’s concern. The primary endpoint of this study is to find out the efficacy of fusion and safety of this procedure. According to previous studies, sex did not affect spine fusion rate, we did not compare it. When analyzing according to sex in this study, there was no difference in fusion rate and fusion grade. (Line 197-198)
“When statistical analysis according to sex was performed, there was no difference in fusion rate and fusion grade.”
Fusion Rates of Different Anterior Grafts in Thoracolumbar Fractures; J Spinal Disord Tech. 2015 Nov;28(9):E528-33
“The sex had no significant influence on fusion, but smoking seemed to influence fusion rates significantly”
Risk Factors for Pseudarthrosis in Minimally-Invasive Transforaminal Lumbar Interbody Fusion; Asian Spine Journal 2018;12(5):830-838.
“Sex, mean age at surgery, and mean follow-up time did not differ between the cohorts”
|
|
Fusion rate 6 month |
Fusion rate 12 month |
Fusion grade 6 month |
Fusion grade 12 month |
|
Sex(p value) |
0.674 |
0.709 |
0.508 |
0.931 |
There is an absence of control subjects and a wide range of SVF cell numbers and cell viability as authors mentioned also force me to suggest the inclusion of more participant before conclude anything.
RESPONSE 4: We appreciate the reviewer’s concern. Although this study was not a study using inter-human comparison, the difference in intra-cage fusion could be observed by dividing the control group and the experimental group between inserted cages. In all 10 patients, β-TCP plus SVF were implanted into the right-side cage, whereas β-TCP only was implanted into the left-sided cage. This allowed intrapatient evaluation of the potential added value of SVF supplementation. (Line 319-323).
We tried to experiment according to the existing number of SVFs as in the proposed reference paper, but there is a variation in the number of SVFs because the amount of fat collected is different for each patient. Also, we used Fresh newly-isolated SVF as one-step as an experiment conducted on humans, it has a relatively wide range of SVF cell numbness and cell viability. We suggested that a large-clinical trial is needed to overcome this point.
Reference:
Autologous mesenchymal stromal cells embedded in tricalcium phosphate for posterolateral spinal fusion: results of a prospective phase I/II clinical trial with long-term follow-up; Stem Cell Res Ther. 2019 Feb 22;10(1):63.
Spinal fusion using adipose stem cells seeded on a radiolucent cage filler: a feasibility study of a single surgical procedure in goats; European Spine Journal volume 24, pages1031–1042(2015)
There is not discussion of including TCP.
RESPONSE 5: Thank you for pointing this out. I added the β-TCP in discussion
“β-TCP is a synthetic bone substitute that combines type 1 collagen and tricalcium phosphate with a highly porous scaffold that supports bone growth. β-TCP is typically augmented by BMA to add osteogenic and osteoinductive qualities.” (Line 244-246)
“SVF has same role with BMAs, so the combination of SVF and β-TCP should have a synergistic effect in our study. Prins HJ et al. demonstrated the feasibility and potential efficacy of SVF seeded on bone substitutes for maxillary sinus floor elevation and autologous SVFs with calcium phosphate ceramics showed increased maxiallry bone height for dental implantations” (Line 292-295)
No data mentioned of VAS in Table 1 except pretreatment VAS score.
RESPONSE 6: We appreciate the reviewer's concern. The pretreatment VAS only mentioned in Table 1 was intended to show the baseline demographic characteristics of the patient. The VAS for pre-treatment and post-treatment are described in “3.2 clinical and radiographic outcomes”. (Line 183-185)
“Pain assessments with VAS were performed preoperatively and at 1, 3, 6, 12 months follow-up. VAS scores were reduced from a mean of 7.9 ± 1.3 preoperatively to 4 ± 0.9 at 1 month, 3.2 ± 1.1 at 3 months, and 2.7 ± 1.6 at 12 months follow-ups, respectively (Table 1). “
Figure legend3- Pls correct the word white arrow in D and E as there is no white arrow. All the MRI and CT images should be labeled properly.
RESPONSE 7: Thank you for pointing this out. We revised the figure and figure legend.
Round 2
Reviewer 2 Report
Dear Authors,
Now the manuscript looks very clear and the main objective focused on very few subjects in the study. There are still some minor changes needed for more clarity.
- Figure 6 needs to be improved to see more visible points.
- Table 1 needs to move after preoperatively (line 184).
- Line 60, replace the word study with studies.